# Insights into the Genomic Potential of a *Methylocystis* sp. from Amazonian Floodplain Sediments

**DOI:** 10.3390/microorganisms10091747

**Published:** 2022-08-30

**Authors:** Júlia B. Gontijo, Fabiana S. Paula, Andressa M. Venturini, Jéssica A. Mandro, Paul L. E. Bodelier, Siu M. Tsai

**Affiliations:** 1Cell and Molecular Biology Laboratory, Center for Nuclear Energy in Agriculture, University of São Paulo (CENA-USP), Piracicaba 13416-000, Brazil; 2Department of Biological Oceanography, Oceanographic Institute, University of São Paulo, São Paulo 05508-120, Brazil; 3Princeton Institute for International and Regional Studies, Princeton University, Princeton, NJ 08544-1019, USA; 4Netherlands Institute of Ecology (NIOO-KNAW), 6708 PB Wageningen, The Netherlands

**Keywords:** MAG, methanotrophs, pan-genome, tropical wetlands, Amazon

## Abstract

Although floodplains are recognized as important sources of methane (CH_4_) in the Amazon basin, little is known about the role of methanotrophs in mitigating CH_4_ emissions in these ecosystems. Our previous data reported the genus *Methylocystis* as one of the most abundant methanotrophs in these floodplain sediments. However, information on the functional potential and life strategies of these organisms living under seasonal flooding is still missing. Here, we described the first metagenome-assembled genome (MAG) of a *Methylocystis* sp. recovered from Amazonian floodplains sediments, and we explored its functional potential and ecological traits through phylogenomic, functional annotation, and pan-genomic approaches. Both phylogenomics and pan-genomics identified the closest placement of the bin.170_fp as *Methylocystis parvus*. As expected for Type II methanotrophs, the Core cluster from the pan-genome comprised genes for CH_4_ oxidation and formaldehyde assimilation through the serine pathway. Furthermore, the complete set of genes related to nitrogen fixation is also present in the Core. Interestingly, the MAG singleton cluster revealed the presence of unique genes related to nitrogen metabolism and cell motility. The study sheds light on the genomic characteristics of a dominant, but as yet unexplored methanotroph from the Amazonian floodplains. By exploring the genomic potential related to resource utilization and motility capability, we expanded our knowledge on the niche breadth of these dominant methanotrophs in the Amazonian floodplains.

## 1. Introduction

The atmospheric concentration of methane (CH_4_) has increased steadily since the l900s [1]. Wetlands are the largest sources of CH_4_, where the gas is generated by methanogenic archaea in the final step of anaerobic degradation of organic matter [2,3]. The Amazonian floodplains, which comprise an area of around 800,000 km^2^ [4], have been suggested to play an important role in global CH_4_ emissions [5,6,7]. Recently, Basso et al. identified a high regional variability of CH_4_ emissions in the Amazon basin [8]. Moreover, the predicted temperature rises due to global climate changes may lead to additional emissions of CH_4_ to the atmosphere [9]. Therefore, it is crucial to understand the factors controlling the flux of CH_4_ in this environment, including the microbes related to CH_4_ consumption, known as methanotrophs.

The CH_4_-oxidizing microbes are traditionally classified into subgroups with distinct ecological niches and aerobic or anaerobic metabolism [10]. Briefly, aerobic methanotrophs are grouped as Type I (from the Gammaproteobacteria class), Type II (from the Alphaproteobacteria class), and Type III (from the Verrucomicrobia phylum) [11,12]. These microbes possess methane monooxygenase (MMO), a key enzyme for CH_4_ oxidation that can be found in both particulate (pMMO, encoded by *pmo* genes) and/or soluble (sMMO, encoded by *mmo* genes) forms [12]. By contrast, anaerobic oxidizing methanotrophs have been described for both bacterial and archaeal taxa, using alternative electron acceptors [13,14,15] or in consortia with sulfate-reducing bacteria [16].

Our previous study reported a high diversity of methanotrophs that use different pathways to oxidize CH_4_ in Amazonian floodplains sediments [17]. The genus *Methylocystis*, in particular, was identified as the most abundant methanotrophic bacterial taxon in those sediments. This Type II methanotroph is among the most abundant and functionally active CH_4_-oxidizing groups in various terrestrial and aquatic environments [12]. It is also known as a non-motile and nitrogen-fixing methanotroph [18]. However, information on the functional potential and life strategies of these organisms living under seasonal flooding tropical regions is still missing. As there is no *Methylocystis* sp. isolated from Amazonian floodplains to date, the use of culture-independent methods based on the reconstruction of microbial genomes from metagenomes may provide insights into the metabolic diversity and lifestyle of these microorganisms. Here, we described the first MAG of *Methylocystis* sp. recovered from Amazonian floodplains sediments. Based on phylogenomics, functional annotation, and pan-genomics, we investigated the potential functional and ecological traits of this MAG, aiming to understand its cellular and metabolic strategies for survival under seasonal flooding conditions.

## 2. Materials and Methods

### 2.1. Sediment Sampling and DNA Extraction

Sediment sampling was performed in two floodplain sites, located on the Amazonas River (FP2, 2°28′11.2″ S 54°38′49.9″ W) and at the intersection between Amazonas and Tapajós Rivers (FP3, 2°22′44.8″ S 54°44′21.1″ W), in the Eastern Brazilian Amazon, Santarém municipality, State of Pará. Triplicate sediment samples were collected from the two sites during the wet (May) and dry (October) seasons of 2016, adding up to a total of 12 samples. DNA was extracted from 0.25 g of sediment by using PowerLyzer PowerSoil DNA Isolation Kit (MoBIO Laboratories Inc., Carlsbad, CA, USA), following the optimized protocol for tropical soils described by Venturini et al. [19]. Further information on the site characteristics, sediment sampling, physicochemical properties, and DNA extraction have been described previously [17].

### 2.2. Metagenomic Sequencing and Bioinformatic Analysis

The metagenomic libraries from the 12 sediment samples were constructed by using the NEBNext Ultra II DNA Library Prep Kit for Illumina (New England Biolabs, Inc., Ipswich, MA, USA), and the paired-end shotgun sequencing (2 × 150 bp) was carried out on an Illumina HiSeq 2500 platform (Illumina, Inc., San Diego, CA, USA) at Novogene Co., Ltd. (Beijing, China). An average of 10 Gb raw data per sample was obtained. Further information regarding the metagenomic sequencing has been previously described by Venturini et al. [20].

The assembly of genomes from the metagenomic data, as well as the analysis of their functional potential, was performed on the KBase 2.1.9 platform [21], and the optimized parameters for each step are described in Appendix A Appendix A. Raw metagenomic sequences were filtered and trimmed by using Trimmomatic 0.36 [22] by removing sequencing adapters, and sequences with lengths lower than 70 bp and Phred score lower than 20. The quality of the sequences was confirmed with FastQC 0.11.5 [23]. After quality control, the paired forward and reverse sequences of the 12 samples were taxonomically classified using Kaiju 1.7.3 [24], and the methanotrophic genera were manually filtered considering the Methanotroph Commons database (http://www.methanotroph.org/wiki/taxonomy/, accessed on 10 May 2022). The same trimmed libraries were merged into a single sequence library by using the app Merge Reads Libraries 1.0.1 [21]. Co-assembly was performed with MEGAHIT 1.2.9 [25]. Binning was performed with contigs > 2000 bp by using MaxBin2 2.2.4 [26] and MetaBAT2 1.7 [27]. DASTool 1.2 [28] was used for consensus binning. CheckM 1.0.18 [29] was used to determine the completeness and contamination of the bins in order to classify them as high-, medium-, or low-quality MAGs, according to the parameters defined by Bowers et al. [30]. Bins with completeness >50% and contamination <10% were taxonomically classified by using GTDB-Tk 1.1.0 [31] and double-checked by using the Microbial Genomes Atlas Online (MiGA) [32].

We further inspected the characteristics of the near-complete bin.170_fp by comparison with eight public genomes of *Methylocystis* species (*M. bryophila, M. heyeri, M. hirsuta*, two *M. parvus*, and three *M. rosea*) available on the NCBI database (Appendix A Appendix A). The reference genomes were selected based on the assembly quality (>95% of completeness and <2% contamination) and species-level classification. These nine genomes were used as inputs for phylogenomic analysis based on a set of 49 universal genes defined by Clusters of Orthologous Groups (COG) gene families (Appendix A Appendix A) by the app Build Microbial SpeciesTree 1.6.0 [21]. The relative abundance of bin.170_fp in relation to its merged sequence library was estimated by using Bowtie2 2.3.2 [33].

We used the software DRAM 0.0.2 [34] for functional annotation. The DRAM output of the bin.170_fp and the public genomes used for phylogenomic analysis were also used as inputs for pan-genomic analysis by using Anvi’o 7 [35], considering the COG annotation [36], and the following parameters: 0.5 for minbit, 1 for the gene cluster minimal occurrence, 2 for MCL inflation, and NCBI blastp for amino acid sequence similarity search. Finally, the predicted proteins from bin.170_fp and the MAG Singletons and Core clusters were compared to sequences in the KEGG database through GhostKOALA [37].

## 3. Results and Discussion

### 3.1. Metagenome-Assembled Genome from Amazonian Floodplain Sediments

From the metagenomic libraries constructed with 12 floodplain sediment samples, we obtained 560,314,186 high-quality reads, which were assembled into 281,029 contigs. The contigs were binned into 45 medium- and high-quality MAGs, assigned to the archaea and bacteria domains (data not shown). Based on the Genome Taxonomy Database (GTDB-tk 1.1.0), bin.170_fp was classified as *Methylocystis* sp., with *Methylocystis parvus* as the closest taxonomic placement, showing relative evolutionary divergence (RED) of 0.986 and the average nucleotide identity (ANI) of 81.41%, indicating that the MAG may represent a novel *Methylocystis* species. Furthermore, similar taxonomic placement was indicated by MiGa, with an average amino-acid identity (AAI) of 72.77%. This MAG was assembled from 266 contigs and contains 3,367,001 bp, 63.48% of GC content, N50 of 18,002, L50 of 51, 96.0% of completeness, and 2.7% of contamination. It is, therefore, considered a near-complete assembly, according to the classification proposed by Parks et al. [38]. Furthermore, Nelson et al. reported that MAGs with more than 90% of completeness and low contamination are likely to represent organismal function effectively [39]. The phylogenomic tree constructed by using the eight *Methylocystis* sp. reference genomes confirmed *Methylocystis parvus* as the closest phylogenetic placement of the MAG recovered from Amazonian floodplains (Figure 1).

*Methylocystis* was one of the first genera of aerobic methanotrophic bacteria described [40]. Some species have been classified as facultative methanotrophs due to their ability to use different carbon substrates, such as methanol and acetate [41]. These Type II methanotrophs are also known to persist in environments under seasonal variations of oxygen (O_2_) and CH_4_ [11], which is likely to confer an advantage in the floodplain environment. The relative abundance of the genus *Methylocystis* in the metagenomic reads ranged from 0.08 to 0.92% (Appendix A Appendix A), whereas the relative abundance of bin.170_fp in the merged metagenomic library used for its assembly was estimated as 0.07%. The importance of this genus in floodplain sediments has also been indicated at the RNA level in a previous study [42], indicating its potential role in mitigating the CH_4_ emissions in this environment.

### 3.2. Functional Annotation and Pan-Genomic Analysis

Bin.170_fp was annotated by using the DRAM pipeline [34]. A total of 3,519 genes were identified as protein-coding genes (CDS), with 2,565 assigned to functions. Appendix A Appendix A presents a summary of the main metabolic routes annotated in bin.170_fp, including the coverage of different metabolic modules and the presence of genes related to carbohydrate-active enzymes (CAZY), CH_4_, and nitrogen (N) metabolisms, as well as the comparison with the reference genomes.

We investigated bin.170_fp using a pan-genomic approach to partition its functional potential among the different gene clusters. A total of 10,949 gene clusters were identified in the *Methylocystis* pan-genome. Based on gene similarity, bin.170_fp presented an ANI of 80% with the closest relative (*M. parvus*). We further examined the Core (genes present in all genomes analyzed) and MAG Singletons (genes present only in bin.170_fp) clusters (Figure 2).

A total of 1,501 gene clusters composed the Core, and 720 gene clusters were classified as MAG Singletons. We investigated the classification of the predicted proteins from the MAG Singletons and the Core cluster, as well as all predicted proteins from bin.170_fp using the KEGG database. Then, we constructed a model based on the predictions for the central metabolism of bin.170_fp and indicated those that were part of the Core and MAG Singletons gene clusters (Figure 3 and Appendix A). As already widely described, the CH_4_ oxidation and formaldehyde assimilation capacities are part of the Core cluster of *Methylocystis* [12]. Regarding the methanotrophic metabolism, only the *pmo*CAB operon was identified in bin.170_fp, whereas no other *mmo* gene was present. In fact, all known *Methylocystis* species have a membrane-bound or particulate methane monooxygenase (pMMO), whereas the presence of the soluble form of this enzyme (sMMO) may vary within the genus [43]. In all methanotrophs that contain the pMMO, copper (Cu) is used as the catalytic metal in the first methanotrophic step, that is, the CH_4_ oxidation to methanol. However, Cu is also known to suppress the expression of sMMO, as reviewed by Guerrero–Cruz et al. [44]. The Amazon river’s waters deposit large amounts of nutrients into its sediments [45], and we previously reported the high contents of Cu in the same sediments studied here [17]. Considering the Amazonian sediments as a non-favorable environment for the sMMO functionality, the presence of this trait would not confer an additional advantage in this habitat, and a pMMO-using pathway is likely to be the dominant route.

Surprisingly, regarding the second methanotrophic step (the oxidation of methanol to formaldehyde), only one gene encoding the methanol dehydrogenase (MDH) was found in bin.170_fp (*xox*F, lanthanide-dependent MDH). Recent studies have shown that *xox*F is widely distributed and may have the same environmental relevance as the conventional calcium-dependent MDH (*mxa*F) [46,47]. The complete set of genes related to the formaldehyde oxidation and assimilation via the serine cycle was also found in the Core cluster, in addition to the ethylmalonyl-CoA pathway for carbon fixation (Figure 3 and Appendix A). Furthermore, bin.170_fp also possesses genes related to acetate assimilation, confirming its potential to grow in different carbon substrates. In fact, the acetate utilization by *Methylocystis* as a multi-carbon source may be important for survival in environments where CH_4_ availability is variable [48], such as in seasonal floodplains.

We also explored functions related to nitrogen metabolism. Genes encoding N fixation capacity were found to be part of the Core cluster. Although the rare *vnf*DKGH (vanadium–iron nitrogenase) may occur in some *Methylocystis* spp. [49], bin.170_fp presented only the classical *nif*DKH (molybdenum–iron nitrogenase) operon. Other N-metabolism-related genes were also identified. However, these genes may not be included in the Core cluster, indicating their presence varies among the different *Methylocystis* species. The complete set of genes related to the assimilatory nitrate reduction (*nas*A and *nir*A), as well as the gene encoding for hydroxylamine oxidoreductase (*hao*), which converts hydroxylamine to nitrite, were found in bin.170_fp. During the assimilatory nitrate reduction, ammonia is formed before being incorporated into amino acids [50]. However, ammonia is known to inhibit methanotrophy via competitive inhibition, as pMMO can also catalyze the oxidation of ammonia to hydroxylamine [51]. This compound is highly toxic; therefore, the *hao* gene can be present in methanotrophs to prevent its accumulation [52,53]. In this regard, Nyerges and Stein found that a *Methylocystis* sp. had a great capacity to detoxify hydroxylamine [54]. Moreover, genes related to dissimilatory nitrate reduction (*nir*B) and denitrification (*nir*K) were also present, and the latter was detected only in bin.170_fp. In fact, in previous studies on genomic analysis, authors have hypothesized the nitrifying and denitrifying capabilities of aerobic methanotrophs [55]. For instance, the presence of genes related to denitrification in *M. hirsuta* CSC1 was also reported by Bordel et al. [56], and the authors suggested a partial denitrification activity under anoxic conditions. The presence of genes related to different N-cycling routes may indicate the potential adaptation of bin.170_fp for oscillating environmental conditions, including N availability and O_2_ limitation that seasonally occur in the Amazonian floodplains.

Lastly, we present interesting results related to the mobility potential of this genome. Bin.170_fp contained a nearly complete set of genes for chemotaxis and flagellar assembly, with some of these genes present exclusively in this MAG when compared to the reference genomes. These included genes encoding for methyl-accepting chemotaxis proteins (MCPs). MCPs, commonly present in motile microbes [57], are responsible for recognizing environmental signals and transmitting this information to *che*A, which controls the direction of flagellar motor rotation [58]. The flagellum is a complex biological nanomachine with more than 60 genes for its formation and is structured in three main parts: the basal body, the hook, and the filament [59]. Figure 4 presents the predicted flagellum assembly of bin.170_fp and highlights the genes shared among the reference genomes and those unique to the MAG. The flagellum can be considered a structural advantage, as the intracellular chemotactic signaling pathways regulate the direction of flagella-driven motility in response to environmental changes, allowing the migration to a more favorable environment for the organism survival [60]. Recently, Oshkin et al. described in *Methylospira mobile*, also a Type II methanotroph isolated from a wetland, the cellular mechanisms and signal transduction proteins that include the flagellar assembly as an adaptation for its lifestyle [61]. However, Oshkin et al. observed that the occurrence of these mobility-related genes is rare among *Methylocystis* spp. and had been detected only in *Methylocystis parvus* strains and *Methylocystis* ATCC 49242 [49]. Therefore, the flagellum presence in bin.170_fp may extend the range of environmental conditions suitable for its niche occupancy in Amazonian floodplains.

Here, we explored the genomic potential of the first MAG identified as *Methylocystis* sp. recovered from Amazonian floodplain sediments. Taken together, our results may indicate the cellular and metabolic adaptations of this methanotroph for surviving in this environment, as the seasonal flooding conditions have an influence on nutrient and O_2_ availability, as well as on microbes’ motion. These different genomic strategies, including genes related to N metabolic pathways as well as motility capacity, may contribute to the niche occupancy of this organism. Such characteristics are essential for understanding the niche breadth of these dominant methanotrophs in the floodplains. Considering the predicted CH_4_ emission stimulation in a warmer scenario and the lack of information about methanotrophs of the Amazon region, including isolates, this study provides progress toward expanding our knowledge about the roles of CH_4_ oxidizing microbes in Amazonian floodplains.

## Figures and Tables

**Figure 1 microorganisms-10-01747-f001:**
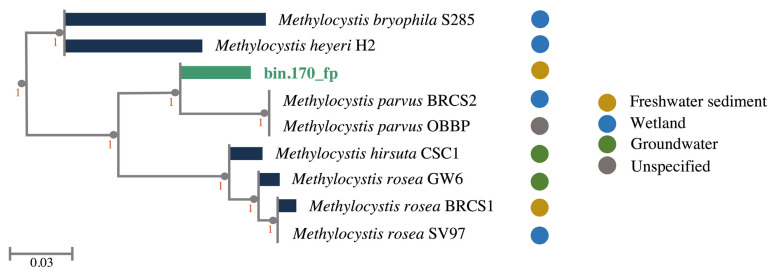
Phylogenomic tree showing the position of bin.170_fp in relation to other *Methylocystis* species. The numbers in the tree are the support values from the Shimodaira–Hasegawa test in the topological split of that branch from the rest of the tree, with 1000 resampling trials (the value 1 is the most confident). Bin.170_fp is represented by the green bar, and the reference genomes are represented by the blue bars. The colored circles represent the environment of isolation or recovery. The root (not shown) is composed of 30 genomes of alphaproteobacterial representatives. The scale bar indicates the number of substitutions per amino acid position.

**Figure 2 microorganisms-10-01747-f002:**
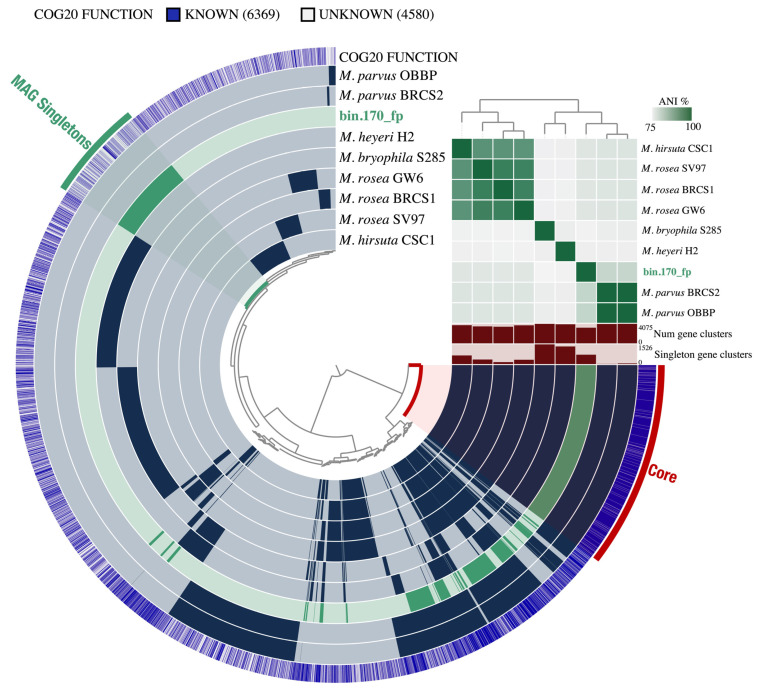
Pan-genome analysis based on the presence/absence of 10,949 gene clusters, which are defined by the tree in the center (Euclidean distance; Ward linkage). Bin.170_fp is represented by the green bar, and the reference genomes are represented by the blue bars. The Core cluster represents the genes present in all genomes analyzed, and the MAG Singletons cluster indicates the genes present only in bin.170_fp. The heatmap represents the ANI percentage between the genomes analyzed (75–100%).

**Figure 3 microorganisms-10-01747-f003:**
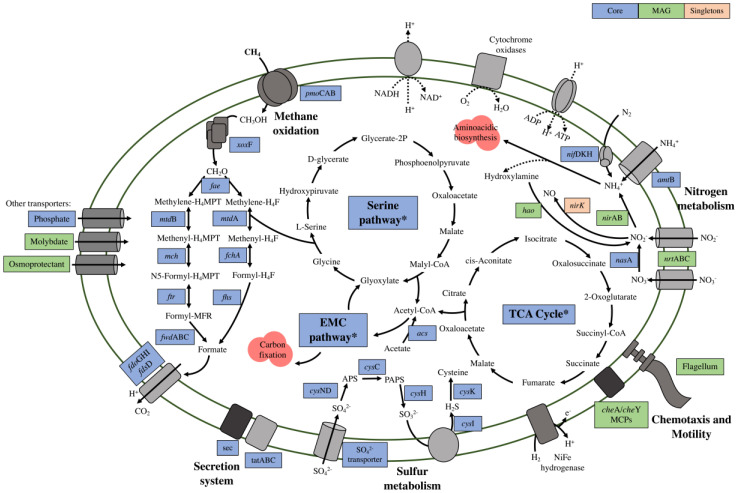
Model prediction for the central metabolism based on the genes’ presence from bin.170_fp MAG compared with the Core and MAG Singletons clusters. The model includes potential capabilities related to CH_4_, nitrogen (N), sulfur (S) metabolisms, as well as the secretion system, chemotaxis, and motility. Genes presented in the Core cluster are represented by the blue squares, and genes presented in bin.170_fp, but not in the Core, are represented by the green squares. Genes indicated by the orange squares are present only in the MAG Singletons cluster. The list of all genes used for the metabolic model prediction (including those from the pathways indicated by *) is available in Appendix A.

**Figure 4 microorganisms-10-01747-f004:**
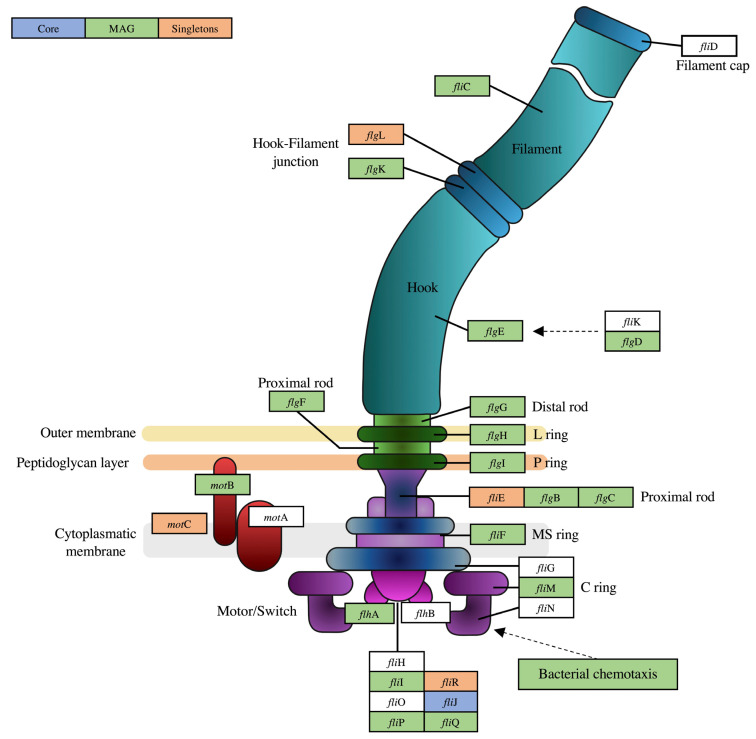
Predicted flagellum assembly for bin.170_fp and comparisons with the Core and MAG Singletons clusters. Genes presented in the Core cluster are represented by the blue squares, and genes presented in bin.170_fp, but not in the Core, are represented by the green squares. Genes indicated by orange squares are present only in the MAG Singletons cluster. Genes indicated by white squares are missing in the assembly. The list of all genes used for the flagellum assembly prediction is available in Appendix A. The model was adapted with permission from Kanehisa et al. [62], 2022, Kanehisa Laboratories.

## Data Availability

The bin.170_fp assembly will be publicly available on GenBank (JAMBEQ000000000) under the umbrella BioProject PRJNA782633 after the acceptance of this manuscript. The 12 raw metagenomic reads are also available under the same BioProject.

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
