# Peer review of "Insights into the Genomic Potential of a Methylocystis sp. from Amazonian Floodplain Sediments"

_microorganisms, 2022, doi:10.3390/microorganisms10091747_

Round 1
Reviewer 1 Report
In this manuscript, the authors report the metagenome-assembled genome of Methylocystis sp. from Amazonian floodplain sediments and investigated its functional and ecological traits through phylogenomic, functional annotation, and pan-genomic approaches. This paper will give new insights into understanding methanotrophic activity in natural environments. However, this manuscript has several issues which should be addressed prior to publication.
Major comments
Formaldehyde assimilation pathway is an important metabolic pathway in methanotrophs. But the authors did not refer to the serine cycle together with carbon fixation pathway in Results and discussion section. The presence of the genes involved in formaldehyde assimilation pathway should be described in the main text.
The formaldehyde oxidation pathway, serine pathway, and carbon fixation pathway are inaccurate in Figure 3. The reference No. 44 is not appropriate as a model because Methylomicrobium sp. is the type I methanotroph which use the ribulose monophosphate pathway for formaldehyde fixation. In the methylotrophs harboring the serine pathway, formaldehyde is oxidized to formate by the H4MPT-dependent oxidation pathway. Key enzyme of this pathway is formaldehyde-activating enzyme (fae) which generates methylene-H4MPT. Do the bin.170_fp and other type strains have a gene for fae? This should be mentioned.
Methylene-H4MPT is not the substrate of serine glyoxylate aminotransferase (sgaA) which generates serine from glycine. This enzyme uses methylene-H4F which is derived from the H4F-dependent formate reduction pathway. For example, please see Fig. 1 in the review by Kalyuzhnaya et al. (Metabolic Engineering 29:142-152, 2015).
The pathway from glycine to “carbon fixation” via tetrahydrofolate in Figure 3 is wrong. Methylotrophs using the serine pathway have the ethylmalonyl-CoA pathway or the glyoxylate pathway for glyoxylate regeneration. For example, Methylocystis parvus OBBP which is the closest relative to the bin.170_fp has enzymes involved in the ethylmalonyl-CoA pathway (Genome Announcement 194:5709-5710, 2012). The authors should discuss the pathway for glyoxylate regeneration. If the bin.170_fp has the ethylmalonyl-CoA pathway, this pathway should be described in Figure 3.
Minor comments
There are many grammatical errors in the whole manuscript. More intensive English check should be done by native speakers. For example,
L120: Should be “data not shown”.
L159-160: Should be rephrased “as well as the and …… genomes”.
L197: Add “as” after “well”.
L225: Should be “the latter”.
Author Response
> Reviewer’s comment:
In this manuscript, the authors report the metagenome-assembled genome of Methylocystis sp. from Amazonian floodplain sediments and investigated its functional and ecological traits through phylogenomic, functional annotation, and pan-genomic approaches. This paper will give new insights into understanding methanotrophic activity in natural environments. However, this manuscript has several issues which should be addressed prior to publication.
>> Answer:
We thank the reviewer for all the insightful comments, which contributed to improve our manuscript. Based on your suggestions, we also updated Figure 3 and included a supplementary table (Table S4) containing a list of all genes used for the metabolic model prediction. Our study provides new insights regarding the dominant aerobic methanotrophs in Amazonian floodplains, and we believe that our results can provide important contributions to the field. We are also very pleased to know that the reviewer agrees with that.
> Reviewer’s comment:
Formaldehyde assimilation pathway is an important metabolic pathway in methanotrophs. But the authors did not refer to the serine cycle together with carbon fixation pathway in Results and discussion section. The presence of the genes involved in formaldehyde assimilation pathway should be described in the main text.
>> Answer:
Thank you for your suggestion. We have included in the Results and Discussion section further information about the formaldehyde assimilation pathway (lines 209-211).
> Reviewer’s comment:
The formaldehyde oxidation pathway, serine pathway, and carbon fixation pathway are inaccurate in Figure 3. The reference No. 44 is not appropriate as a model because Methylomicrobium sp. is the type I methanotroph which use the ribulose monophosphate pathway for formaldehyde fixation. In the methylotrophs harboring the serine pathway, formaldehyde is oxidized to formate by the H4MPT-dependent oxidation pathway. Key enzyme of this pathway is formaldehyde-activating enzyme (fae) which generates methylene-H4MPT. Do the bin.170_fp and other type strains have a gene for fae? This should be mentioned.
>> Answer:
Initially, we had cited the reference 44, but in fact they are methanotrophs of different metabolisms, so we removed this reference. Furthermore, the bin.170_fp carries the fae gene. The information regarding the formaldehyde oxidation were included in the main text (lines 209-211), Figure 3 and Table S4.
> Reviewer’s comment:
Methylene-H4MPT is not the substrate of serine glyoxylate aminotransferase (sgaA) which generates serine from glycine. This enzyme uses methylene-H4F which is derived from the H4F-dependent formate reduction pathway. For example, please see Fig. 1 in the review by Kalyuzhnaya et al. (Metabolic Engineering 29:142-152, 2015).
>> Answer:
Thanks for pointing it out. We have updated the Figure 3 and now the information regarding this pathway are accurate.
> Reviewer’s comment:
The pathway from glycine to “carbon fixation” via tetrahydrofolate in Figure 3 is wrong. Methylotrophs using the serine pathway have the ethylmalonyl-CoA pathway or the glyoxylate pathway for glyoxylate regeneration. For example, Methylocystis parvus OBBP which is the closest relative to the bin.170_fp has enzymes involved in the ethylmalonyl-CoA pathway (Genome Announcement 194:5709-5710, 2012). The authors should discuss the pathway for glyoxylate regeneration. If the bin.170_fp has the ethylmalonyl-CoA pathway, this pathway should be described in Figure 3.
>> Answer:
As mentioned in Methods section, based on the pangenomic analysis the predicted proteins of the selected gene clusters were annotated by GhostKOALA (KEGG). On this platform, it is possible to visualize the metabolic maps, as the map00680 (genome.jp/pathway/map00680), which is related to the methane metabolism, that indicates the “carbon fixation” via tetrahydrofolate. Indeed, this is a very simplistic representation of this pathway. We agree with your suggestion to include in the Results and Discussion section and in the Figure 3 the ethylmalonyl-CoA pathway for glyoxylate regeneration based on the map00630 (genome.jp/pathway/map00680), which is complete in the bin.170_fp and also in the core cluster. Thanks for pointing this out.
> Reviewer’s comment:
There are many grammatical errors in the whole manuscript. More intensive English check should be done by native speakers. For example,
L120: Should be “data not shown”.
L159-160: Should be rephrased “as well as the and …… genomes”.
L197: Add “as” after “well”.
L225: Should be “the latter”.
>> Answer:
We have double checked the English and corrected grammar errors. Thank you for pointing it out.
Reviewer 2 Report
The authors attempt to make a case for a genomic insight into, what they presume, a dominant methanotroph in the site of the study, part of the Amazon floodplain. However, from the data presented, it is just a regular Methylocystis type that is omnipresent across a variety of environments and across different climatic zones. So, this MAG that represents this dominant organism does not appear to present much novelty or any new insight into methanotrophy, either in the Amazon or elsewhere. Unless the authors could provide a good argument for why this single genome sequence, very similar to the previously published genome sequences is special or insightful, I only see this work as a genome announcement. While motility may not be typical of some Methylocystis species, it has been identified in the close relatives of the organism reported here. While the authors find it surprising that only the XoxF-type methanol dehydrogenase was detected, it is a well known fact that XoxF methanol dehydrogenase are much more persistent in the methylotrophs than the MxaFI type methanol dehydrogenases. To conclude, I have no issues with the validity of the data presented, only with the novelty of the content. Overall, I do not think any insight has been gained about how the function of Methylocystis in the Amazon is different from the function of Methylocystis in a Norwegian bog. They seem to be just the same.
Author Response
> Reviewer’s comment:
The authors attempt to make a case for a genomic insight into, what they presume, a dominant methanotroph in the site of the study, part of the Amazon floodplain. However, from the data presented, it is just a regular Methylocystis type that is omnipresent across a variety of environments and across different climatic zones. So, this MAG that represents this dominant organism does not appear to present much novelty or any new insight into methanotrophy, either in the Amazon or elsewhere. Unless the authors could provide a good argument for why this single genome sequence, very similar to the previously published genome sequences is special or insightful, I only see this work as a genome announcement. While motility may not be typical of some Methylocystis species, it has been identified in the close relatives of the organism reported here. While the authors find it surprising that only the XoxF-type methanol dehydrogenase was detected, it is a well known fact that XoxF methanol dehydrogenase are much more persistent in the methylotrophs than the MxaFI type methanol dehydrogenases. To conclude, I have no issues with the validity of the data presented, only with the novelty of the content. Overall, I do not think any insight has been gained about how the function of Methylocystis in the Amazon is different from the function of Methylocystis in a Norwegian bog. They seem to be just the same.
>> Answer:
We thank the reviewer for the insightful comments. As the reviewer stated, Methylocystis is a widespread and well-studied genus. It has been described in several environments, what highlights its ecological importance. We appreciate the concerns regarding the novelty of the study. Although the genome presented contains some particularities, pointed out in the pangenome analysis as the singletons, including in functions related to the nitrogen cycle and motility, it indeed contains many similarities with published genomes.
Because of the significant contribution of the Amazonian floodplains for the regional and global methane cycle, several ongoing studies are aiming to understand the controls of methane cycle in this region. In a recent study, Basso et al. (2021) reported an atypical elevated CH4 emission in Santarém region, the same region of our study, what reinforces the importance of the study in this area.
Our manuscript is part of a bigger project, where we are aiming to identify the microbial groups underpinning the CH4 cycling in eastern Amazon wetlands, and Methylocystis is believed to play an important role in CH4 oxidation. We believe the study of the characteristics of this organism’s genomes and the discussion of its ecology and adaptations provides a very valuable resource for the scientific community. The analysis and discussion presented are much more insightful than a simple genome announcement publication, mainly because studies investigating genomic details for organisms retrieved from this area are scarse. Therefore, I believe this justifies the importance of the study.
Round 2
Reviewer 2 Report
The contents of the manuscript have not changed after revision. The novelty remains low. The authors argue otherwise. I have no further comments. I leave it up to the editors to accept or to reject the manuscript.